Microbiology
Spectrum
# Impact of Temperature and Oxygen Availability on Gene Expression Patterns of *Mycobacterium ulcerans*

Laxmi Dhungel,ᵃ Raisa Bonner,ᵃ Meagan Cook,ᵃ Duncan Henson,ᵃ Trent Moulder,ᵃ M. Eric Benbow,ᵇ,ᶜ,ᵈ,ᵉ 🅳 Heather Jordanᵃ

ᵃDepartment of Biological Sciences, Mississippi State University, Mississippi State, Mississippi, USA
ᵇDepartment of Entomology, Michigan State University, East Lansing, Michigan, USA
ᶜEcology, Evolution and Behavior Program, Michigan State University, East Lansing, Michigan, USA
ᵈAgBioResearch, Michigan State University, East Lansing, Michigan, USA
ᵉDepartment of Osteopathic Medical Specialties, Michigan State University, East Lansing, Michigan, USA

**ABSTRACT** Buruli ulcer disease is a neglected tropical disease caused by the environmental pathogen *Mycobacterium ulcerans*. The *M. ulcerans* major virulence factor is mycolactone, a lipid cytotoxic compound whose genes are carried on a plasmid. Although an exact reservoir and mode(s) of transmission are unknown, data provide evidence of both. First, Buruli ulcer incidence and *M. ulcerans* presence have been linked to slow-moving water with low oxygen. *M. ulcerans* has also been suggested to be sensitive to UV due to termination in *crtI*, encoding a phytoene dehydrogenase, required for carotenoid production. Further, *M. ulcerans* has been shown to cause disease following puncture but not when introduced to open abrasion sites, suggesting that puncture is necessary for transmission and pathology. Despite these findings, the function and modulation of mycolactone and other genes in response to dynamic abiotic conditions such as UV, temperature, and oxygen have not been shown. In this study, we investigated modulation of mycolactone and other genes on exposure to changing UV and oxygen microenvironmental conditions. Mycolactone expression was downregulated on exposure to the single stress high temperature and did not change significantly with exposure to UV; however, it was upregulated when exposed to microaerophilic conditions. Mycolactone expression was downregulated under combined stresses of high temperature and low oxygen, but there was upregulation of several stress response genes. Taken together, results suggest that temperature shapes *M. ulcerans* metabolic response more so than UV exposure or oxygen requirements. These data help to define the environmental niche of *M. ulcerans* and metabolic responses during initial human infection.

**IMPORTANCE** Buruli ulcer is a debilitating skin disease caused by the environmental pathogen *Mycobacterium ulcerans*. *M. ulcerans* produces a toxic compound, mycolactone, which leads to tissue necrosis and ulceration. Barriers to preventing Buruli ulcer include an incomplete understanding of *M. ulcerans* reservoirs, how the pathogen is transmitted, and under what circumstances mycolactone and other *M. ulcerans* genes are expressed and produced in its natural environment and in the host. We conducted a study to investigate *M. ulcerans* gene expression under several individual or combined abiotic conditions. Our data showed that mycolactone expression was downregulated under combined stresses of high temperature and low oxygen but there was upregulation of several stress response genes. These data are among only a few studies measuring modulation of mycolactone and other *M. ulcerans* genes that could be involved in pathogen fitness in its natural environment and virulence while within the host.

**KEYWORDS** Buruli ulcer, *Mycobacterium ulcerans*, mycolactone, gene expression

Address correspondence to Heather Jordan, jordan@biology.msstate.edu.

The authors declare no conflict of interest.

10.1128/spectrum.04968-22 **1**

Buruli ulcer is a neglected tropical disease caused by an environmental pathogen, *Mycobacterium ulcerans*. The disease is characterized by an often-painless nodule that can later develop into an ulcer (1). Buruli ulcer is the third most common mycobacterial infection and has been reported in over 33 countries worldwide (1). Buruli ulcer is often associated with functional limitations and limb deformities in cases of deferred treatment and imposes a significantly negative medical, psychological, and socioeconomic impact on affected patients (2, 3). The major virulence factor of *M. ulcerans* is mycolactone, a lipid cytotoxic compound whose genes are carried on the large plasmid pMUM001 (4). Mycolactone diffuses through healthy tissue, leading to pathology that impacts a wide variety of cells through modulation of immune functioning or inhibiting immune-regulating proteins or by causing cytoskeletal rearrangement, cell cycle arrest, necrosis, or apoptosis, depending on cell type (5).

Morbidity has, in part, been exacerbated by the fact that the mode of *M. ulcerans* transmission remains unknown; however, multiple mechanisms have been proposed and investigated (6, 7). Epidemiological evidence links Buruli ulcer incidence to slow-moving aquatic habitats (8–10). This has also been supported by the finding of *M. ulcerans* DNA in association with aquatic plant biofilms, water filtrand, soil, and invertebrates (11–15). Slow-moving water with low oxygen concentration has also been included as a risk factor (9, 10, 15–17). Laboratory studies showed *M. ulcerans* leads to disease pathology following puncture but not when introduced to an open abrasion site (18, 19), and mosquitoes and other biting insects have also been implicated as potential mechanical vectors (7).

As an environmental pathogen, *M. ulcerans* naturally resides in a complex community of aquatic life presumably structured by biological interactions and abiotic environmental factors (6). In reviewing other systems, such as the environmental pathogens *Burkholderia cenocepacia* and *Vibrio vulnificus*, low oxygen concentration and high temperature have been shown to upregulate virulence factors (20, 21). Additionally, temperature-regulated toxin production has been observed in pathogenic bacteria such as enterohemorrhagic *Escherichia coli* (EHEC), *Yersinia enterocolitica*, and *Bacillus anthracis* (22). Data from these other pathogens' studies suggest a need to investigate the possible role of higher temperature and lower oxygen yielding changes in mycolactone production, in establishment of disease and fitness within its natural environment (19). Understanding modulation of mycolactone and other gene expression in response to temperature and oxygen gradients will aid not only in understanding *M. ulcerans* response to changing abiotic conditions in aquatic environments but also in understanding *M. ulcerans* virulence and pathogenesis leading to disease, depending upon transmission route.

*M. ulcerans* is also suggested to be UV sensitive due to termination in *crtL*, a gene responsible for carotenoid production (23). As a replicative reservoir of *M. ulcerans* is unknown within aquatic habitats, it is not clear whether *M. ulcerans* replicates in areas that are protected from UV or has developed machineries to counteract adverse UV effects (6). Thus, investigating whether mycolactone influences protection against UV through a pigment-mediated or other mechanism can provide further insight into *M. ulcerans* reservoirs.

Few studies have been published exploring mycolactone gene expression. One study showed that transcription of several key mycolactone biosynthetic genes is driven by a SigA-like promoter (24); however, the study did not determine environmental or growth-phase signals inducing mycolactone gene expression. An *in vitro* study showed mycolactone genes downregulated in response to various sugar sources (25), and other research showed that nutrient availability (chitin versus calcium) regulates several metabolic pathways in *M. ulcerans*; however, mycolactone toxin was not expressed in these nutrient-abundant environments, suggesting that its expression may be regulated mainly under stressful conditions (26).

In this study, we investigated how exposure to single and combined abiotic factors affected *M. ulcerans* growth, modulation of mycolactone expression via real-time quantitative PCR (RT-qPCR), and global gene expression of a subset of samples via transcriptome

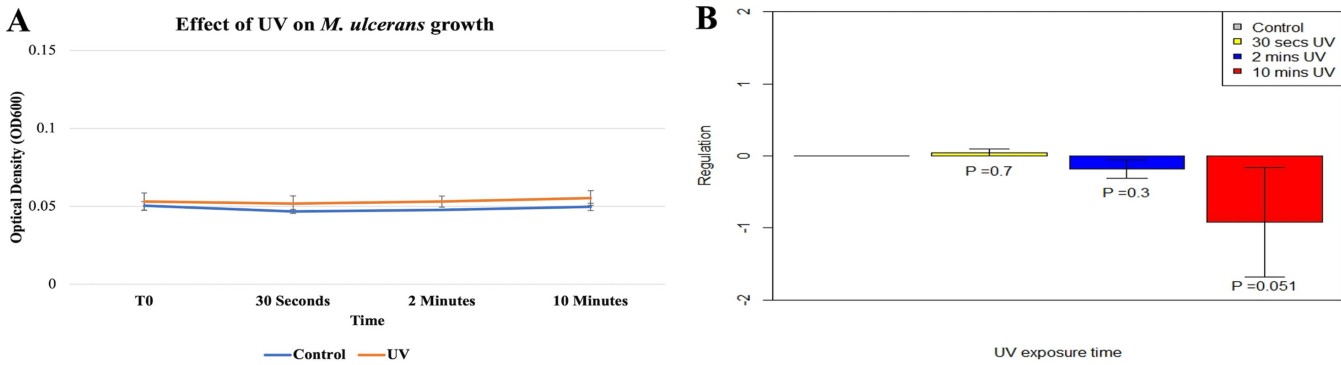

**FIG 1** Effects of increasing UV exposure on *M. ulcerans* growth (A) and ER expression (B).

sequencing (RNA-Seq). First, we tested the effect of UV exposure on *M. ulcerans* growth and mycolactone gene expression and then whether growth and mycolactone and other *M. ulcerans* genes were modulated under changing temperature and oxygen, and the combined interactive effect of the latter two. These data fill gaps in knowledge regarding *M. ulcerans* stress response, providing molecular evidence for how fitness may be influenced by abiotic conditions of aquatic ecosystems and in Buruli ulcer wounds.

## RESULTS

**Little effect of UV exposure on *M. ulcerans* growth and ER gene expression.** Compared to controls, no significant difference in *M. ulcerans* growth was observed following exposure to UV for 30 s ($P = 0.18$), 2 min ($P = 0.45$), or 10 min ($P = 0.10$) (Fig. 1A). There was a negligible effect on mycolactone gene expression upon exposure of *M. ulcerans* to UV for 30 s, whereas there was a slight downregulation with exposure to UV for 2 min, but neither of these effects was statistically significant (Fig. 1B). However, enoyl reductase (ER) gene expression trended toward significant downregulation on exposure to UV for 10 min ($P = 0.051$, Fig. 1B).

**_M. ulcerans_ growth and mycolactone expression when exposed to single and combined environmental stressors.** Across all time points there was no statistical difference in *M. ulcerans* growth (at 30°C) under microaerophilic or anaerobic conditions in comparison to aerobic conditions (Fig. 2A). Similarly, there was no significant difference in *M. ulcerans* growth when subjected to different temperatures (30°C versus 37°C) for 24 h (Fig. 2B). When cultures were brought back to 30°C, those that had been subjected to 37°C were not significantly different from those subjected to 30°C based on growth (Fig. 2B).

Exposure of *M. ulcerans* to microaerophilic conditions for 24 h showed significant upregulation of ER gene expression ($P = 0.0009$). Gene expression was slightly downregulated when *M. ulcerans* exposed to microaerophilic conditions at 30°C was transferred back to aerobic conditions, but the difference was not statistically significant (Fig. 3). *M. ulcerans* exposed to anaerobic conditions for 24 h had nonsignificant upregulation of mycolactone gene expression. However, upon transferring the anaerobic-condition-exposed bacteria back to aerobic conditions, there was significant ($P = 0.005$) upregulation of mycolactone gene expression compared to controls grown under aerobic conditions over the 3-day study (Fig. 3).

ER gene expression was significantly downregulated when *M. ulcerans* grown aerobically at 30°C (control) was exposed aerobically at 37°C ($P = 0.02923$) for 24 h and significantly upregulated when brought back to aerobic conditions at 30°C ($P = 0.0002$ compared to control and $P = 0.001$ compared to 37°C under aerobic condition-day 2). Although ER gene expression was downregulated when the bacteria were exposed to 37°C under microaerophilic conditions and was upregulated when the bacteria were

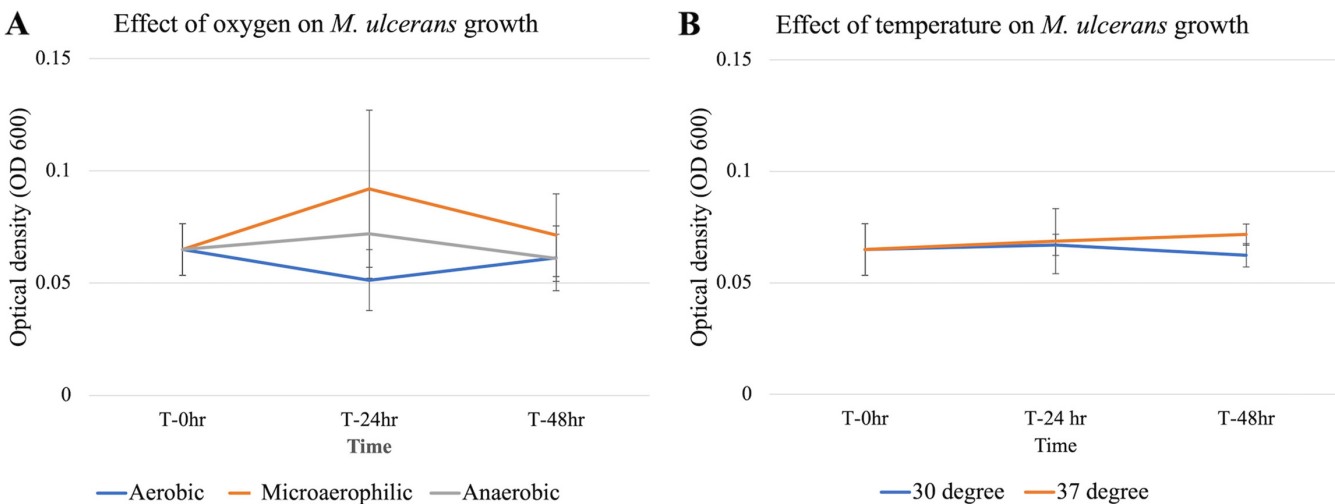

**FIG 2** Effect of oxygen (A) or temperature (B) on *M. ulcerans* growth. (A) Optical density of *M. ulcerans* when exposed to aerobic (blue line), microaerophilic (orange line), and anaerobic (gray line) conditions at 30°C. Exponential *M. ulcerans* initially under aerobic conditions was exposed to its respective oxygen condition for 24 h and then transferred back to aerobic conditions for an additional 24 h (48 h from initial time point). (B) Optical density of *M. ulcerans* when exposed to 30°C (blue line) and 37°C (orange line). Exponential *M. ulcerans* initially at 30°C was exposed to its respective temperature conditions for 24 h and then exposed again to 30°C for an additional 24 h (48 h from initial time point).

brought back to 30°C and aerobic conditions, the regulation was not statistically significant (Fig. 4A).

The shift of *M. ulcerans* at 30°C under aerobic conditions to 37°C and anaerobic conditions showed significant downregulation of ER gene expression ($P = 0.002$). When the cultures were brought back to control conditions, ER gene expression was not statistically different from the control but was significantly upregulated ($P = 0.01$) compared to *M. ulcerans* at 37°C under anaerobic conditions on day 2 (Fig. 4B).

**M. ulcerans global stress response on exposure to high-temperature and low-oxygen conditions.** RNA-Seq analysis was performed to determine regulated gene response to exponentially grown *M. ulcerans* at 30°C under aerobic conditions with exposure to 37°C or to the combination of 37°C and microaerophilic conditions. A heat map representing 50 of the top significant differentially regulated genes across treatments and time points compared to the control condition is shown in Fig. 5; however, a heat map and list of all significantly differentially regulated genes can be found in Fig. S1 and Table S1, respectively, in the supplemental material.

RNA-Seq data showed 187 differentially regulated genes when *M. ulcerans* was grown aerobically at 37°C compared to those that remained under control conditions for the duration of the experiment. There were 158 upregulated genes including one gene involved in Environmental Informational Processing, 19 with Genetic Information Processing, 45 with Metabolism, and 4 with Signaling and Cellular Processes. Two genes were uncharacterized, and 87 had no KEGG orthology (KO) assigned. Twenty-nine genes were significantly downregulated compared to *M. ulcerans* under control conditions, including 3 involved in Environmental Informational Processing, 4 with Genetic Information Processing, 13 involved in Metabolism (3 of which were MUP001 plasmid genes MUP032c, MUP039c, and MUP040c, encoding MLSB, MLSA2, and MLSA1, respectively), and 2 involved in Signaling and Cellular Processes (4, 24). Seven significantly downregulated genes had no KO assigned (Fig. 5, Fig. S1, and Table S1).

*M. ulcerans* moved back to control conditions on day 3 from being grown aerobically at 37°C showed only 12 significantly upregulated genes compared to *M. ulcerans* grown under control conditions for the entire experiment. These included 5 genes involved in Genetic Information Processing, 6 genes involved in Metabolism (including MUP032c and MUP039c), and 1 with no KO assigned. Only seven genes were significantly downregulated compared to those under control conditions, including 4 involved in Metabolism and 3 with no KO assigned (Fig. 5, Fig. S1, and Table S1).

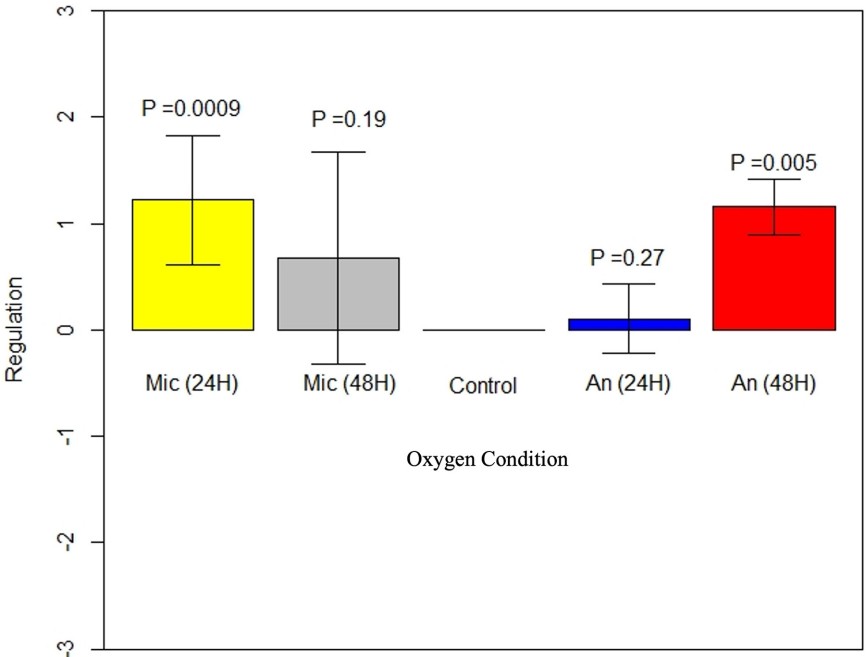

**FIG 3** *M. ulcerans* ER regulation after exposure to microaerophilic or anaerobic conditions compared to aerobic conditions. Exposure to a microaerophilic environment for 24 h caused significant upregulation [Mic (24H); *P* = 0.0009] of ER expression (yellow bar) and transferring the bacteria back to aerobic conditions led to a slight downregulation in ER expression [Mic (48H), gray bar]. Exposure of *M. ulcerans* to anaerobic conditions for 24 h led to slight ER gene upregulation [AN (24H), blue bar], but transfer back to aerobic conditions caused significant ER upregulation [AN (48H), *P* = 0.005, red bar] compared to control *M. ulcerans* exposed to aerobic conditions during the entire 3-day experiment (Control bar). Error bars indicate standard errors. *M. ulcerans* was at 30°C for all the oxygen conditions.

When *M. ulcerans* was grown microaerophilically at 37°C, there were 372 significantly and differentially regulated genes compared to *M. ulcerans* grown under control conditions for the entire experiment. A total of 349 genes were upregulated, including 3 genes involved in Environmental Informational Processing, 38 involved in Genetic Information Processing (including MUP001 plasmid gene MUP005c, encoding a possible chromosome partitioning protein, ParA) (27), 110 with Metabolism, and 7 with Signaling and Cellular Processes. Three genes were uncharacterized, and 188 had no KO assigned. Twenty-three genes were significantly downregulated compared to control conditions, including 2 involved in Environmental Informational Processing, 2 with Genetic Information Processing, 11 involved in Metabolism (3 of which were MUP001 plasmid genes MUP039c and MUP040c, encoding MLSA2 and MLSA1, respectively), and 1 involved in Signaling and Cellular Processes. Seven significantly downregulated genes had no KO assigned.

When *M. ulcerans* was returned to control conditions on day 3 after being grown microaerophilically at 37°C, there were only 2 significantly upregulated genes compared to *M. ulcerans* grown under control conditions for the entire experiment. These upregulated genes included one gene involved in Metabolism, and 1 with no KO assigned. Only 5 genes were significantly downregulated compared to control, with all 5 being involved in Metabolism, including MUP032c.

Finally, comparison of gene expression levels among *M. ulcerans* growing at 37°C either aerobically or microaerophilically showed 76 genes that were upregulated under the microaerophilic conditions compared to aerobic condition. These included 12 genes involved in Genetic Information Processing, 28 genes involved in Metabolism, 4 involved with Signaling and Cellular Processes, and 32 with no KO assigned. There were no statistically significant differences in downregulated *M. ulcerans* genes between either of the oxygen treatment conditions with growth at 37°C.

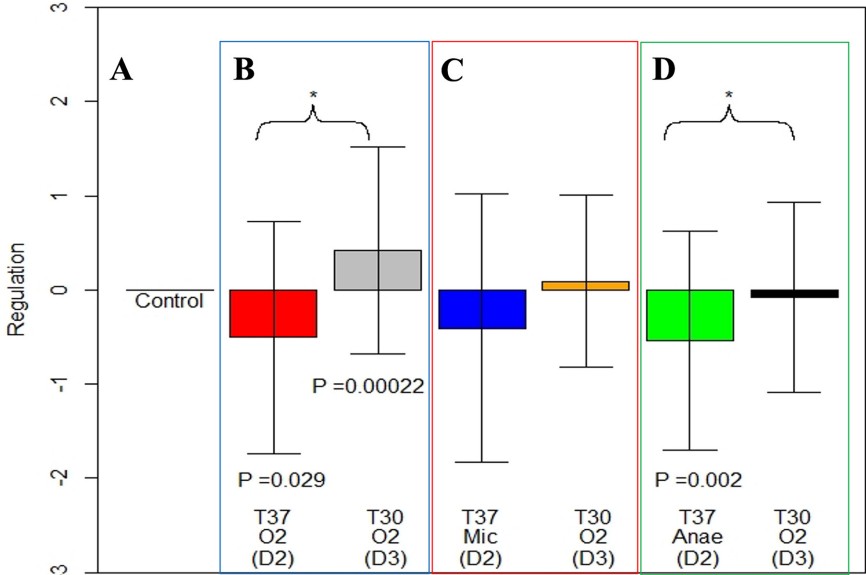

**FIG 4** Regulation of ER expression after *M. ulcerans* exposure to differing temperature and oxygen conditions. (A) Control conditions of *M. ulcerans* at 30°C and aerobic conditions from days 1 to 3; (B and in blue box) *M. ulcerans* exposed to 37°C and aerobic conditions on day 2 [T37 O2 (D2)] but transferred back to 30°C and aerobic condition on day 3 [T30 O2 (D3)]; (C and in red box) *M. ulcerans* exposed to 37°C and microaerophilic condition on day 2 [T37 Mic (D2)] but transferred back to 30°C and aerobic condition on day 3 [T30 O2 (D3)]; (D and in green box) *M. ulcerans* exposed to 37°C and anaerobic condition on day 2 [T37 Anae (D2)] but transferred back to 30°C and aerobic condition on day 3 [T37 O2 (D3)]. Bars indicate standard error. *P* values indicate significance values between treatment and control. Asterisks indicate significance within treatments.

## DISCUSSION

The capacity of *Mycobacterium ulcerans* to sense, respond to, and adapt to variable and hostile environmental conditions inevitably makes it successful in its natural environment and increases its ability to survive in its host. With this in mind, we conducted one of the first studies investigating *M. ulcerans* growth and transcriptional response (including modulation of mycolactone gene expression and global responses) to the abiotic conditions of UV, temperature, and oxygen, in an effort to better understand the *M. ulcerans* environmental niche and factors promoting *M. ulcerans* pathogenesis.

Premature termination of the *crtI* gene in *M. ulcerans* that protects its progenitor, *Mycobacterium marinum*, against sunlight damage suggests either that *M. ulcerans* resides in UV-protected areas or there is a presence of other machineries to counteract the damage (28, 29). However, in our study, *M. ulcerans* growth was not affected by UV radiation exposure for 10 min. Further investigations exposing *M. ulcerans* to longer durations of radiation are needed to confirm its resistance to UV radiation over longer exposure times. Although our study showed downregulation of mycolactone (ER) expression on UV exposure via RT-qPCR, this was not significant. Under laboratory conditions, wild-type *M. ulcerans* produces bright-yellow-pigmented colonies, while mycolactone mutants are white, suggesting mycolactone-mediated protection may be possible (4, 25). In our study, the color of UV-treated *M. ulcerans* colonies was yellow on M7H10 agar plates (data not shown), indicating mycolactone production. Quantitation of mycolactone production and the use of a mycolactone-negative mutant will further elucidate the effect of UV beyond gene expression. Further studies on transcriptional analysis are also required to understand regulation of other machineries that can confer photoprotection and/or DNA repair to protect *M. ulcerans* against UV damage.

Exposure to higher-temperature and lower-than-optimal-oxygen conditions is also known to modulate stress response and virulence genes in environmental pathogens

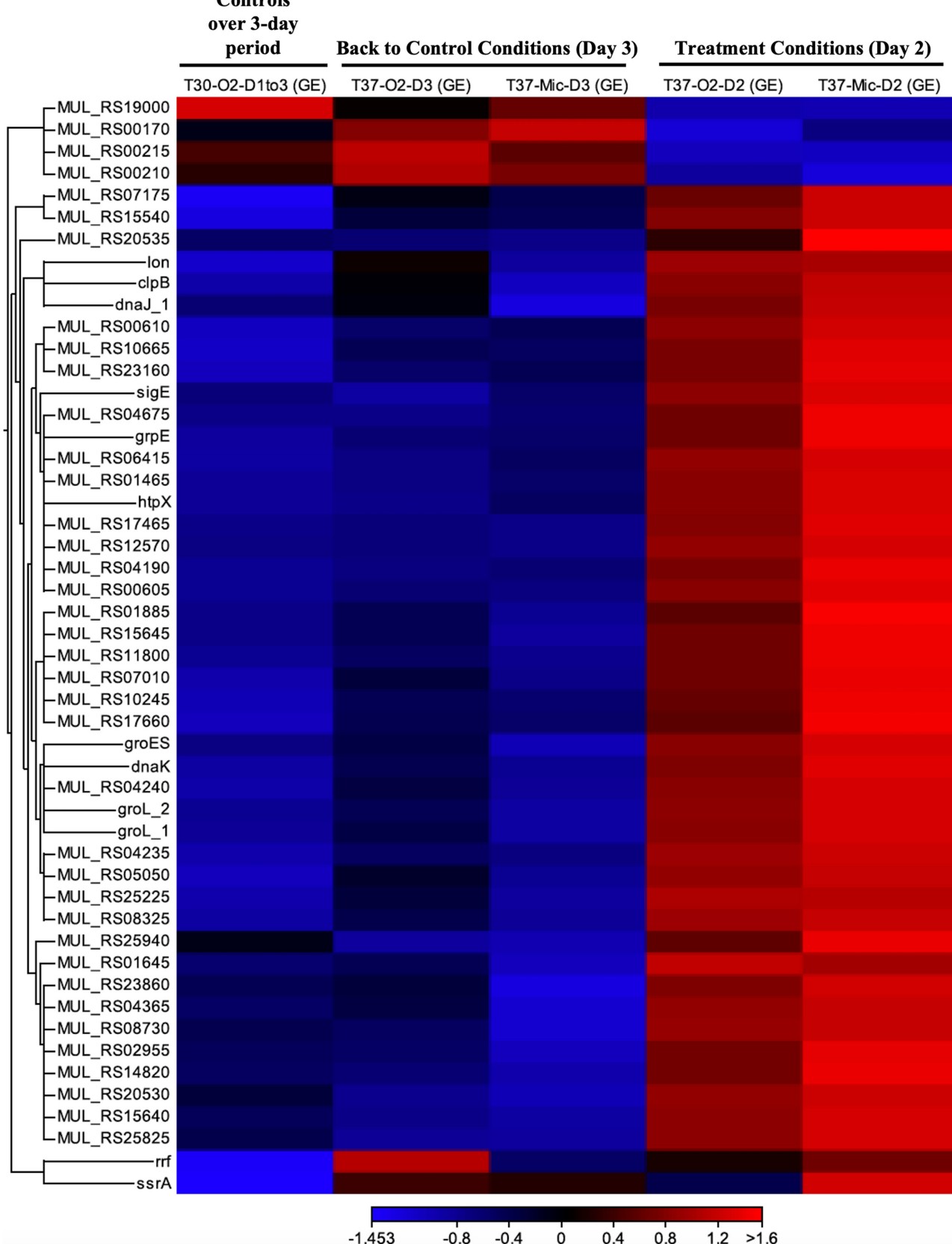

**FIG 5** Fifty significantly differentially regulated genes across treatments and time points. T30-O2-D1to3, *M. ulcerans* grown aerobically at 30°C for the entire 3-day study; T37-O2-D3, *M. ulcerans* grown aerobically at 37°C for 24 h but then moved back to control conditions for 24 h; T37-Mic-D3,

that can aid in the bacterial survival and pathogenicity inside the human host (30, 31). Differences in oxygen conditions did not impact *M. ulcerans* growth in our study. *M. ulcerans* exposure to anaerobic conditions caused slight upregulation of ER gene expression, which, however, was statistically nonsignificant. But when *M. ulcerans* was brought back to aerobic conditions 24 h later, ER gene expression was significantly up-regulated. The combining of anaerobic and 37°C conditions caused significant down-regulation of the ER gene, which was significantly upregulated when brought back to aerobic and 30°C conditions. *M. ulcerans* lacks anaerobic pathway genes and because of this is suggested to be adapted to an aerobic or microaerophilic environmental niche (28). However, its association with mammalian feces and asymptomatic gut colonization raises questions regarding viability yet lack of pathogenicity in the anaerobic intestinal environment (32–37). Although the single-stress anaerobic condition did not affect mycolactone gene expression, combining high temperature and anaerobic conditions downregulated mycolactone gene expression in our study and may account for the lack of pathogenesis in the mammalian intestine. Indeed, in *Staphylococcus aureus*, another pathogen system, toxic shock syndrome toxin 1 (TSST-1) was produced in a $CO_2$ environment (38, 39), increasing production when there was a shift from anaerobic conditions to increasing oxygen concentrations (2%); however, toxin production was decreased for increasing oxygen concentration (6%) in an environment without $CO_2$ (39). Similarly, a slight reduction in oxygen concentrations in microaerophilic environments could enhance ER gene expression, but this effect may not occur at 0% oxygen in anaerobic environments, suggesting the possibility of a very narrow niche of *M. ulcerans* and mycolactone expression.

Under the 30°C-microaerophilic condition, the ER gene was significantly upregulated, but it was downregulated when brought back to aerobic conditions. On the other hand, ER was significantly downregulated when *M. ulcerans* was placed under 37°C-aerobic conditions and then significantly upregulated when brought back to 30°C-aerobic conditions. Further, our *M. ulcerans* RNA-Seq data from the 37°C-aerobic treatment showed downregulation of *mlsA1*, *mlsA2*, and *mlsB*, type I polyketide synthase genes whose proteins synthesize the upper side chain and mycolactone core and the acyl side chain, respectively (4, 24). Exposing *M. ulcerans* to combined micro-aerophilic and 37°C stress conditions led to ER gene downregulation that was not statistically significant through RT-qPCR, though *M. ulcerans* RNA-Seq data under these combined conditions showed significant downregulation of *mlsA1* and *mlsA2*. These data contrast with our initial hypothesis that predicted upregulation of mycolactone in response to combined effects of high-temperature and low-oxygen conditions, which we presumed to be similar in the subdermal environment during human infection.

When considering how our *in vitro* data could fit within the infection model, one must consider the skin environment. The partial pressure of oxygen ($PO_2$) is lower at different layers of skin (superficial, $8.0 \pm 3.2$ mm Hg; dermal papillae, $24.0 \pm 6.4$; and subpapillary plexus, $35.2 \pm 8.0$) than $PO_2$ at atmosphere (160 mm Hg) (40). The sudden exposure to higher-temperature and lower-oxygen conditions depending on the skin layer could induce adaptive stress response mechanisms in *M. ulcerans*, causing it to use its energy efficiently for production of enzymes and other proteins and lipids. Triggering appropriate responses that allow survival and propagation under these conditions could compromise mycolactone synthesis, while also directing the expression of other genes modulating virulence and pathogenicity. However, these are speculations that require much deeper examination for mechanistic validation.

Nevertheless, the combined stress of 37°C-microaerophilic conditions upregulated genes for lipid metabolism (acyl coenzyme A [acyl-CoA] dehydrogenases) and mycolic acid synthesis (*accD6*, *mas*, and *ethA*), similar to what has been shown in *Mycobacterium bovis* and *Mycobacterium tuberculosis* (41–43). Another upregulated gene, *eth*, encodes

**FIG 5** Legend (Continued)

*M. ulcerans* grown microaerophilically at 37°C for 24 h but then moved back to control conditions for 24 h; T37-O2-D2, *M. ulcerans* grown aerobically at 37°C for 24 h; T37-Mic-D2, *M. ulcerans* grown microaerophilically at 37°C for 24 h. All genes listed in the heat map are labeled as listed in RefSeq for a particular gene or locus tag given in the reference genomes' annotation. GE, gene expression.

oxidoreductase that oxidizes keto-mycolic acid to synthesize waxy mycolic acids (44). The upregulation of these genes in our study suggests that combined temperature-oxygen stress caused disturbance in overall *M. ulcerans* metabolism and cell wall synthesis. Upregulation of the *mmsA* gene, which generates propionyl-CoA that produces fatty acids required for cell envelope formation, suggests effects of combined stress on the cell membrane (45–47).

Genes encoding transcriptional regulators SigB, SigE, and WhiB5 were also among significantly upregulated genes in *M. ulcerans* compared to controls within the 37°C-microaerophilic treatment. Sigma B is one of the principal sigma factors and is considered a general stress responder in mycobacteria (48, 49). SigB also positively regulates expression of chaperonins such as *groEL2* and *groES*, antigens such as ESAT-6-like proteins, and cell wall-associated and lipid metabolism-related genes in *M. tuberculosis* (50). WhiB proteins are redox-sensing transcriptional regulators (51). WhiB5 positively regulates 58 genes including type VII secretion systems (ESX-2 and -4) (51). WhiB5 proteins in *M. tuberculosis* are relatively stable and underexpressed under aerobic conditions but are slightly upregulated at 0% oxygen (52). They are suggested to be immunomodulators and enhancers of *M. tuberculosis* survival during nutrient limitation (51).

Finally, stress response genes *hsp20*, *grpE*, *clpB*, *groES*, *groEL1* and *groEL2*, *dnaJ*, *dnaK*, *ahpC*, and *ahpD* were also among those upregulated on exposure of *M. ulcerans* under combined 37°C-microaerophilic conditions. These genes respond to heat shock and oxidative stress (41, 49, 53). The *clpB* gene is associated with virulence in several Gram-positive (e.g., *S. aureus*) and Gram-negative (e.g., *Salmonella enterica* serovar Typhimurium) pathogens (54). In *M. tuberculosis*, *clpB* enhances biofilm formation and promotes survival against hypoxia and heat stress and inside macrophages (54). Similarly, *dna*K provides *M. tuberculosis* protection against heat shock and oxidative stress inside the macrophage (43, 55). Proteins ClpB, GroES, GroEL1, DnaK, AhpC, and AhpD are found in the extracellular matrix (ECM) of *M. ulcerans* biofilm, and *M. ulcerans* with ECM have higher colonization and virulence (56). Upregulation of these *M. ulcerans* genes *in vitro* suggests that environmental signals such as temperature, oxygen, UV, pH, etc., could trigger adaptive responses of *M. ulcerans* to these stresses during infection or within polymicrobial communities in its natural environment. These genes could promote biofilm formation and enhance colonization and virulence activity of *M. ulcerans*. And, although mycolactone is considered a major virulence factor, more investigations are needed to understand the impact of genes (and their products) such as *clpB* and others that are known virulence determinants in other bacteria. Another important consideration is that in this study we defined higher-temperature and lower-than-optimal-oxygen conditions as "stressed conditions" for *M. ulcerans* based on current knowledge about its growth under lab conditions; however, these environmental conditions may not be a "stressed environment" for *M. ulcerans* in its natural habitat and its response to fluctuating temperature and oxygen conditions may simply be a "response" instead of a "stress response."

Many of the same genes discussed above that were upregulated in the 37°C-microaerophilic treatment were also upregulated in the 37°C-aerobic treatment. But interestingly, no statistical differences in downregulated genes were found when comparing the combined stresses of 37°C and microaerophilic treatments to the single stress of 37°C-aerobic treatment. Altogether, our data suggest a higher impact of temperature than oxygen on *M. ulcerans* and mycolactone expression. One possible explanation may be that the microaerophilic conditions induce upregulation of mycolactone expression, but the combined stress of high temperature and low oxygen prioritizes the expression of other essential genes (i.e., stress response and fatty acid degradation genes as discussed above), thereby compromising mycolactone synthesis. This is a mechanism found in other pathogens. For instance, in *Vibrio cholerae*, the promoter of cholera toxin regulator (*toxR*) and heat shock (*htpG*) genes overlap but are transcribed in opposite directions (57, 58). Hence, an increase in temperature allows transcription of the *htpG* gene with reduction in *toxR* expression. Another consideration is that, in *M. ulcerans*, the

ER domain is present repeatedly in the *mlsA* gene (3 times in *mlsA1* and 1 time in *mlsA2*); however, this domain is absent in the *mlsB* gene (4). Hence, ER represents one of many possible genes involved in mycolactone synthesis, but its regulation does not solely depict mycolactone gene expression. Thus, this may account for the discrepancy in RT-qPCR and RNA-Seq results for mycolactone expression.

Finally, data support that mycolactone-producing mycobacteria (MPM) such as *M. ulcerans* and other ecological variants evolved from a common *M. marinum* progenitor by undergoing various gene deletions and pseudogene formation and rearrangement and acquiring plasmid pMUM to adapt to a specific ecological niche (59). In the evolutionary hierarchy, *Mycobacterium liflandii* is suggested to be an intermediate between the ancestor *M. marinum* M and *M. ulcerans Agy*99, as *M. marinum* consists of all gene clusters present in *M. liflandii* and *M. liflandii* consists of all gene clusters present in *M. ulcerans* (60). But there are some genes that are pseudogenized in *M. ulcerans* but not in *M. liflandii* and vice versa, indicating that there was a significant and independent reductive evolution of their genomes. These differences in mutation patterns along with the variation in type of mycolactone produced indicate that these variants experience different sets of environmental pressures and have adapted to occupy different niches, underscored by differing hosts (6, 61). Therefore, it would be interesting to determine whether these responses are also observed for other MPM.

In conclusion, *M. ulcerans* acquired plasmid pMUM001 at the expense of a large deletion in its genome, thereby suggesting its specific role in adaptation to a particular environment (59). In this study, *M. ulcerans* was exposed to several abiotic stresses to understand their effect on *M. ulcerans* growth and mycolactone and other gene expression. Our data suggest that *M. ulcerans* may reside in a microaerophilic habitat in the environment and mycolactone could provide a fitness advantage in those environments. Data also suggest that *M. ulcerans* may have higher tolerance to UV than previously thought and mycolactone may provide a fitness advantage in this context. The exposure of *M. ulcerans* to combined high temperature (37°C) and low oxygen upregulated several stress response genes and other genes known to be involved in virulence in *M. tuberculosis*, while mycolactone gene expression was downregulated, suggesting other virulence factors may be utilized by *M. ulcerans*. Further, downregulation of mycolactone expression on exposure to combined anaerobic and 37°C stress could partly explain the reason behind the asymptomatic gut colonization of different mammals as reported in previous studies (35). Some limitations to this study, such as small sample size, short duration of exposure of *M. ulcerans* to UV radiation (up to 10 min), and no transcriptome analysis to study other machineries that can protect *M. ulcerans* against UV damage, should be addressed in future studies. Additionally, short time points were selected for this study to understand the impacts of short-term exposure on *M. ulcerans* response through changes in gene expression; however, measuring responses to longer exposure times will be important in future experiments. Finally, investigations measuring mycolactone production and using a mycolactone-negative *M. ulcerans* mutant and other MPM are needed to further elucidate the role of mycolactone against these and other abiotic stresses. But altogether, these initial data increase our understanding of *M. ulcerans* response to a changing environment and open doors to future studies that may provide insight into the *M. ulcerans* environment and pathogenesis upon host infection.

## MATERIALS AND METHODS

**Bacterial strains and culture.** A 1% inoculum of *Mycobacterium ulcerans* JKD8083 or *Agy*99 was inoculated into a 30-mL total volume of Middlebrook 7H9 (M7H9) broth containing oleic acid-albumin-dextrose-catalase (OADC) and incubated aerobically at 30°C for 4 to 6 weeks to reach exponential phase for use in this study.

**Measurement of optical density at 600 nm (OD$_{600}$).** *M. ulcerans* cells form aggregates in culture. Hence, aggregates were broken by passage through a 20G syringe, followed by a 25G syringe 10 times. Optical density was measured using a ThermoScientific Genesys 20 spectrophotometer, with M7H9 medium used as a blank. Syringe passage of *M. ulcerans* was used for all experiments.

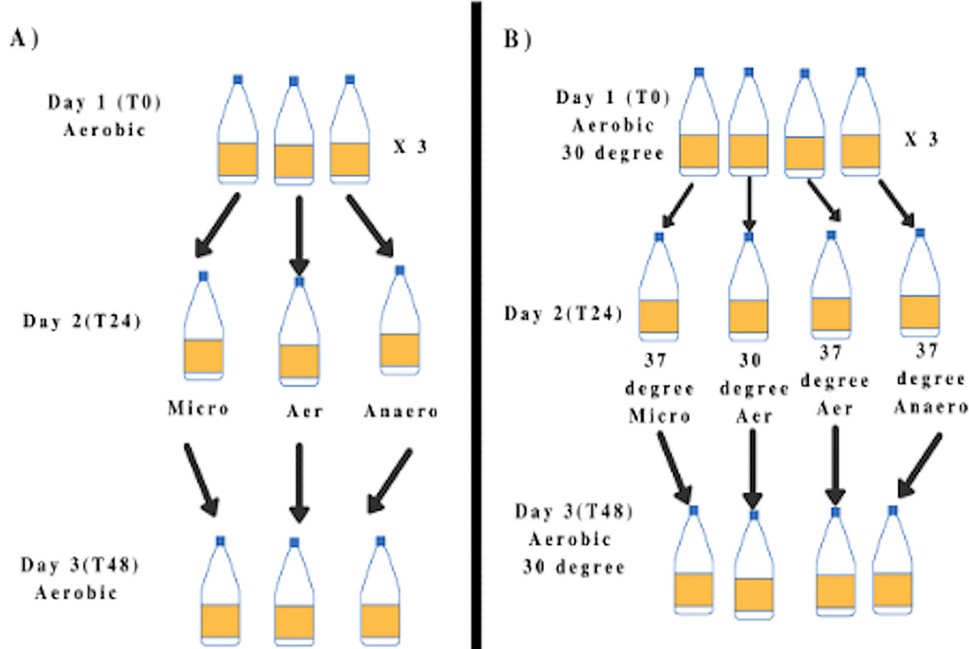

**FIG 6** Schematic representation of study on the effect of oxygen (A) and combined high-temperature (37°C) and low-oxygen (microaerophilic and anaerobic) condition (B) on mycolactone and *M. ulcerans* global gene expression.

**Measurement of bacterial growth.** *M. ulcerans* was serially diluted in $\log_{10}$ concentrations (from undiluted to $10^{-6}$) and plated onto M7H10 agar plates (10 $\mu$L) in triplicate using the spread plate technique and incubated at 30°C. Colonies were counted to determine CFU per milliliter after 4 to 6 weeks to determine the effect of the abiotic factors on *M. ulcerans* growth, and a mean was taken for the triplicates, with standard errors calculated.

**Effect of UV exposure on *M. ulcerans* growth and mycolactone gene expression.** Mycolactone expression was measured on exposure to UV at different time intervals (30 s, 2 min, and 10 min). The time for UV exposure was determined based on work by David et al., where *M. tuberculosis* and *M. marinum* were exposed to UV for up to 30 s and were inactivated (90%) by 7 s and 22 s of UV exposure, respectively (62). *M. ulcerans* during exponential growth ($5.7 \times 10^5$ CFU/mL, 30 mL) was transferred to individual petri plates and exposed to UV (254-nm wavelength) for the respective time interval to measure *M. ulcerans* growth and modulation of mycolactone gene expression. Controls included *M. ulcerans* transferred to petri plates but without UV exposure. At each time point and condition, 1 mL of *M. ulcerans* was transferred for serial dilution and growth measurement by optical density and CFU count, and 5.0 mL of sample was transferred for RNA isolation and RT-qPCR to measure mycolactone gene expression as described below.

**Effect of temperature and oxygen on *M. ulcerans* growth and mycolactone gene expression.** To determine the effects of different oxygen conditions on *M. ulcerans* growth, *M. ulcerans* initially grown aerobically at 30°C was exposed to either microaerophilic or anaerobic conditions at 30°C for 24 h, by placing those cultures within anaerobic chambers with appropriate GasPaks (BD) and oxygen indicators (Fig. 6A). Optical density was measured initially and after 24 h. *M. ulcerans* cultures were then placed back under aerobic conditions at 30°C for an additional 24 h (48 h from initial time point). The optical density for each treatment and time point was measured. To study the effects of temperature, *M. ulcerans* in exponential growth aerobically at 30°C was exposed to 37°C for 24 h and then placed back in the 30°C incubator for an additional 24 h. The optical density was measured for each treatment and time point. At each time point and condition, 1 mL of sample was transferred for serial dilution and growth measurement, and 5.0 mL of sample was transferred for RNA isolation and RT-qPCR to measure mycolactone gene expression. Each experiment was conducted with triplicate replicates and at least three times.

**Combined effect of high temperature and low oxygen in mycolactone and global gene expression.** Exponentially grown *M. ulcerans* (4 to 6 weeks) aerobically and at 30°C was exposed to 37°C under microaerophilic or anaerobic conditions for 24 h (Fig. 6B). After 24 h, the cultures were again brought back to control conditions (30°C and aerobic) for an additional 24 h (48 h from initial time point). Optical densities were measured for each condition and time point, with samples also collected and analyzed as described above. Each experiment was conducted with triplicate replicates (Fig. 6B).

**RNA isolation.** *M. ulcerans* RNA was isolated using the TRIzol method, according to the manufacturer's instructions. Bacterial cells were pelleted by centrifugation, with supernatant removed, and

1.0 mL TRIzol reagent was added to the pellet and mixed thoroughly and bead beaten. After incubation for 1 h, chloroform was added and centrifuged for phase separation. The aqueous phase containing RNA was obtained and precipitated using isopropanol followed by washing with 75% ethanol. The pellet was dried and dissolved in nuclease-free water to obtain RNA suspension. RNA was quantified using the Qubit 2.0 fluorometer, integrity was verified by gel electrophoresis, and the RNA was DNase treated as necessary.

**Preparation of cDNA.** cDNA was prepared with appropriate controls using the Verso cDNA synthesis kit according to the manufacturer's instructions. The reaction mixture included 4 $\mu$L synthesis buffer, 2 $\mu$L deoxynucleoside triphosphate (dNTP) mix, 1 $\mu$L random hexamer, 1 $\mu$L Verso enzyme, and 1 $\mu$L reverse transcription (RT) enhancer (to prevent genomic DNA carryover); the mixture was added to the template and heated at 42°C for 1 h to obtain cDNA. The absence of genomic DNA was confirmed by qPCR.

**RT-qPCR.** Quantitative real-time PCR (RT-qPCR) targeting the enoyl reductase (ER) domain of module B of pMUM001 responsible for mycolactone production was performed on cDNA (15, 63, 64). The polyphosphate kinase (*ppk*) gene was used as a reference gene (25). The master mix contained 1 $\mu$L of each forward and reverse primer for the *ppk* gene and ER gene, 2.5 $\mu$L of ER probe and *ppk* probe, 12.5 $\mu$L of master mix, 0.5 $\mu$L water, and 3 $\mu$L template cDNA per well of PCR plate. The forward primer for ER was 5′-CGCCTACATCGCTTTGG-3′, and the reverse primer was 5′-ATTGAATCGCAGCCATACC-3′. The forward *ppk* primer was 5′-CGGGAAACTACAACAGCAAGACC-3′, and the reverse *ppk* primer was 5′-CCACCA ACAGATTGCGATAGG-3′. PCR was conducted on triplicate samples using a Bio-Rad CFX96 with parameters that included 95.0°C for 10:00 min, and 39 cycles of 95.0°C for 15 s, 55.0°C for 30 s, and 57.0°C for 30 s.

**RNA-Seq analysis.** RNA libraries were created from combined triplicate replicates of *M. ulcerans* RNA samples under aerobic and 30°C (Day 1-Day 3), aerobic and 37°C (Day 2), and microaerophilic and 37°C (Day 2) conditions, and samples that were transferred from 37°C, aerobic, or microaerophilic conditions to aerobic and 30°C at 48 h (aerobic and 37°C [Day 3], microaerophilic and 37°C [Day 3]) conditions. Libraries were created using the NEBNext Ultra RNA library prep kit and NEBNext Multiplex Oligos (dual index primers) for Illumina and associated protocols. High-throughput RNA sequencing was performed by St. Jude Children's Research Hospital on an Illumina HiSeq2000 with 2- by 150-bp PE (paired-end) read lengths. Sequences were initially trimmed by the sequencing facility using TrimGalore v0.4.2, but a more stringent quality trimming was also performed using default parameters within the Qiagen CLC Workbench 20.0.1 (https://digitalinsights.qiagen.com/) following quality control (QC) analysis of sequence reads. *M. ulcerans* Agy99 and plasmid pMUM001 reference genomes were joined (reference sequence NC_008611 joined with reference sequence NC_005916, assembly GCF_000013925.1, https://www.ncbi.nlm.nih.gov/assembly/GCF_000013925.1), and RNA-Seq data were mapped with the following parameters: (i) maximum number of allowed mismatches was set at 2, with insertions and deletions set at 3; (ii) length and similarity fractions were set to 0.9, with autodetection for both strands; (iii) minimum number of hits per read was set to 10. All genes listed in heat maps are labeled as listed in RefSeq for a particular locus tag given in the reference genomes' annotation.

Differential expression was measured in the CLC Workbench that used the assumption that transcripts with similar average expression levels had similar variability, according to the CLC manual. Statistical differential expression tests were performed based on a negative binomial generalized linear model similar to that of edgeR (65). Differentially expressed genes were generated based on a false-discovery-rate (FDR)-corrected *P* value using the Wald test for comparing the effects of treatments compared to control, or treatments compared across time. Treatment reads with a fold change of 1.5 or higher and an FDR-adjusted *P* value less than or equal to 0.05 were considered significant (66). Statistically significant, differentially regulated gene transcripts were further annotated into pathways by linking protein identifier (ID) with potential conserved domains and protein classifications archived within the Conserved Domain Database (Conserved Domains and Protein Classification), and by using the UniProt (UniProt), KEGG (Kanehisa Laboratories), and STRING (ELIXIR Core Data Resources) databases, gene annotations within NCBI (https://www.ncbi.nlm.nih.gov/nuccore/NC_008611.1 and https://www.ncbi.nlm.nih.gov/nuccore/NC_005916), mycolactone locus patent information (https://www.freepatentsonline.com/y2006/0024806.html), and the Mycobrowser genomic and proteomic database for reference against other mycobacterial species (https://mycobrowser.epfl.ch/genes/Rv0753c).

**Statistical analysis.** Significant difference changes in *M. ulcerans* growth under control conditions compared to abiotic treatment conditions were determined using a Student *t* test. RT-qPCR data were analyzed by relative quantification of gene expression compared to the control using python code implementing the threshold cycle ($\Delta\Delta C_T$) method (67). The reference gene used was *ppk*, and ER was used for the target gene. The fold change in gene expression was determined to obtain regulation relative to control (baseline). If the fold change relative to control was greater than 1, then the gene was considered upregulated. If fold change was less than 1, then it was considered downregulated. The amount of downregulation (for fold change of 0 to 1) was determined by calculating the negative of the reciprocal of fold change, as described by Babu (68). The significant cutoff value ($\alpha$) for upregulation and downregulation was *P* = 0.050.

**Data availability.** Raw sequences are archived in the NCBI Sequence Read Archive (SRA) under BioProject accession number PRJNA907849.

## SUPPLEMENTAL MATERIAL

Supplemental material is available online only.

**SUPPLEMENTAL FILE 1**, XLSX file, 0.1 MB.

**SUPPLEMENTAL FILE 2**, EPS file, 2.3 MB.
**SUPPLEMENTAL FILE 3**, PDF file, 1 MB.

## ACKNOWLEDGMENTS

This work was supported, in part, by the joint NSF-NIH-NIFA Ecology and Evolution of Infectious Disease program (DEB 1911457) awarded to H.J. and M.E.B. The funders had no role in study design, data collection and interpretation, or the decision to submit the work for publication.

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
