## [Reviewer comments · Microbiology Spectrum]

Microbiology Spectrum

Impact of Temperature and Oxygen Availability on Gene Expression Patterns of *Mycobacterium ulcerans*

Laxmi Dhungel, Raisa Bonner, Meagan Cook, Duncan Henson, Trent Moulder, Eric Benbow, and Heather Jordan

Corresponding Author(s): Heather Jordan, Mississippi State University

Review Timeline:

Submission Date:	December 2, 2022
Editorial Decision:	January 3, 2023
Revision Received:	January 18, 2023
Accepted:	January 29, 2023

Editor: John Attack

Reviewer(s): Disclosure of reviewer identity is with reference to reviewer comments included in decision letter(s). The following individuals involved in review of your submission have agreed to reveal their identity: Adewale Adegboyega Oke (Reviewer #3)

Transaction Report:

DOI: <https://doi.org/10.1128/spectrum.04968-22>

January 3, 2023

Dr. Heather Jordan
Mississippi State University
Biological Sciences
P.O. Box GY
Mississippi State, Mississippi 39762

Re: Spectrum04968-22 (Impact of Temperature and Oxygen Availability on Gene Expression Patterns of *Mycobacterium ulcerans*)

Dear Dr. Heather Jordan:

Thank you for submitting your manuscript to Microbiology Spectrum. As you will see your paper is very close to acceptance. Please modify the manuscript along the lines I have recommended. As these revisions are quite minor, I expect that you should be able to turn in the revised paper in less than 30 days, if not sooner. If your manuscript was reviewed, you will find the reviewers' comments below.

When submitting the revised version of your paper, please provide (1) point-by-point responses to the issues raised by the reviewers as file type "Response to Reviewers," not in your cover letter, and (2) a PDF file that indicates the changes from the original submission (by highlighting or underlining the changes) as file type "Marked Up Manuscript - For Review Only". Please use this link to submit your revised manuscript. Detailed instructions on submitting your revised paper are below.

Link Not Available

Sincerely,

John Attack

Reviewer comments:

Reviewer #3 (Comments for the Author):

I consider this research as novel considering the findings which can serve as scientific platform for future work towards better understanding of the pathogenesis of the bacterium with potential stimulation for better diagnoses and management of Buruli ulcer disease. These research findings are good for future. My suggestion is to extend this research looking for possible other virulence factors.

Preparing Revision Guidelines

Please return the manuscript within 60 days; if you cannot complete the modification within this time period, please contact me. If you do not wish to modify the manuscript and prefer to submit it to another journal, please notify me of your decision immediately so that the manuscript may be formally withdrawn from consideration by Microbiology Spectrum.

Dhungel, et al. Reviewer Comments:

- Fig. 1 and 2** Why was *M. ulcerans* growth measured differently for the UV exposure experiment compared to temperature and oxygen level experiments? Should use optical density to measure growth after UV exposure. If you cannot do this for some reason (for example, because of uneven exposure of the UV rays to bacteria in liquid media), you should explain this in your text.
- Line 131** Is $p=0.051$ statistically significant? "P-value less than or equal to 0.05 were considered significant"
- Line 133** Should address the growth rate of *M. ulcerans*. Is 24 or even 48 hours long enough to really measure an effect on growth? Why were timepoints for experiments chosen? These seem like short time points to measure *M. ulcerans* growth. You should justify why you selected these timepoints.
- Line 136** Should mention the temperature used for the experiment represented with Figure 2A. I think it is (30°C) but I am not sure. Same with Figure 3.
- Line 140** Why is 25°C not mentioned? It is shown in figure 2B.
- Line 151** Why is Mic (48H) shown before Mic (24H) on Figure 3. Should be reversed.
- Line 159** I find the labeling on Figure 4 confusing. I suggest a less complicated labeling system. The descriptions are good, but the actual labels confused me.
- Line 188** I was confused by the labeling and categories on Figure 5. Do the last two categories represent a different time frame from the first three? Why do they not represent a 3-day period? The D2 categories should be compared to the control at D2 not the control at D3 and other categories at D3.
- Line 426-427** How many CFUs (or volume of bacterial suspension) were initially inoculated into how many mLs of M7H9?
- Line 433** What spectrophotometer was used?
- Line 435** When was optical density used to measure *M. ulcerans* growth and when were CFU counts on serial dilutions plated on M7H10 agar used? I am not sure if both techniques were used for each experiment, but this should be clarified. (This comment corresponds to the first comment).
- Line 436** How many log 10 dilutions were done, and how many mLs of each dilution was plated?
- Line 445** What is meant by "Briefly"?
- Line 446** How was exponential growth determined?
- Line 480** Was the supernatant removed before Trizol addition?

Line 492 Figure 6 seems to be misplaced. Should be under section starting at line 454.

Microbiology Spectrum

Impact of Temperature and Oxygen Availability on Gene Expression Patterns of *Mycobacterium ulcerans*

Confidential remarks for the Editors

I have taken adequate time to review this manuscript and can clearly state that the conclusions are in concordance with the data presented by the authors. The English and grammar in the entire manuscript reflect proper sentences that are written in a standard correct manner for easy understanding. Very well-designed study and written nicely.

The research work is very interesting and novel as it addresses the gene expression among *M. ulcerans* as related to the bacterium's adaptability to harsh conditions vis-a-vis the virulence factor mycolactone involvement. Experimental methods are adequate. The authors have given scientific platform for future work towards better understanding of the pathogenesis of the bacterium with potential stimulation for better diagnoses and management of Buruli ulcer disease.

This is a good manuscript suitable for publication. There is no need for revision.

I hereby recommend it for acceptance. The Manuscript should be accepted.

Comments and Suggestions for the Author

I consider this research as novel considering the findings which can serve as scientific platform for future work towards better understanding of the pathogenesis of the bacterium with potential stimulation for better diagnoses and management of Buruli ulcer disease.

These research findings are good for future. My suggestion is to extend this research looking for possible other virulence factors.

We thank the reviewers for their diligent review, suggests for edits and future directions. We have addressed each point below and have highlighted edited text in the “Marked up Manuscript” pdf. Specific responses to comments are:

Reviewer 3

Reviewer Comments: I consider this research as novel considering the findings which can serve as scientific platform for future work towards better understanding of the pathogenesis of the bacterium with potential stimulation for better diagnoses and management of Buruli ulcer disease.

These research findings are good for future. My suggestion is to extend this research looking for possible other virulence factors.

Author Response: We thank the reviewer for such positive comments and agree one hundred percent that this is a wonderful starting place to measure other potential virulence factors, particularly in an *in vivo* model. We hope to be able to start conducting this work using an *invivo* model this year as we have received some internal funding to do so.

Reviewer 2

Reviewer Comment: Fig. 1 and 2 Why was *M. ulcerans* growth measured differently for the UV exposure experiment compared to temperature and oxygen level experiments? Should use optical density to measure growth after UV exposure. If you cannot do this for some reason (for example, because of uneven exposure of the UV rays to bacteria in liquid media), you should explain this in your text.

Author Response: Author Response: We collected both sets of data (optical density and plated dilutions for CFUs/mL) at the same time that other samples and data were collected (at the specific timepoints). We reported CFU data when available, but there were some instances where our plates for CFU counts were unreadable due to fungal contamination of some of the triplicate plates used for growth analyses, giving incomplete data across some of the timepoints. In those cases, we instead reported optical density data. However, based on your suggestion, we have changed out Figure 1A to report OD600 data rather than CFU, for consistency across figures. The updated figure is:

Figure 1. Effects of increasing UV exposure on *M. ulcerans* growth (A) and ER expression (B).

Reviewer Comment: Line 131 Is $p=0.051$ statistically significant? “P-value less than or equal to 0.05 were considered significant”

Author Response: We consider that any value greater than 0.05 as not being significant. However, we have edited the last sentence in the “Statistical analysis” sub section of the Methods to read: The significant cut-off value (α) for upregulation and downregulation was $P=0.050$ to clarify this.

Reviewer Comment: Line 133 Should address the growth rate of *M. ulcerans*. Is 24 or even 48 hours long enough to really measure an effect on growth? Why were timepoints for experiments chosen? These seem likeshort time points to measure *M. ulcerans* growth. You should justify why you selected these timepoints.

Author Response: While we did not expect to see major changes in increased growth (around 1.5 doubling over the 2 days), we wanted to collect and report these data to verify that there would not be significant changes in cell density of the treatments compared to controls that might arise as a result of exposure to these “stressors” that could have suggested to account for the changes in gene expression that we were measuring. However, measuring gene expression was our primary goal. The short time points were selected to understand the impacts of short term exposure on *M. ulcerans* response through changes in gene expression; however, we agree with the reviewer that measuring responses to longer exposure times will be important experiments to be conducted as followup. We have edited the last paragraph of the manuscript to include the need for investigations of longer exposure times.

Reviewer Comment: Line 136 Should mention the temperature used for the experiment represented with Figure 2A. I think it is (30°C) but I am not sure. Same with Figure 3.

Author Response: We have included the temperature information (30°C) within the body of the manuscript by editing the sentence “Across all time points there was no statistical difference in *M. ulcerans* growth (at 30°C) under microaerophilic or anaerobic conditions in comparison to aerobic conditions (Figure 2A)”. We have also edited the legend for Figure 2 to read: “...A) Optical density of *M. ulcerans* when exposed to aerobic (blue line), microaerophilic (orange) and anaerobic (grey line) conditions at 30°C...”. Also, we have edited the sentence “Gene expression was slightly downregulated when *M. ulcerans* exposed to microaerophilic conditions at 30°C were transferred back to aerobic conditions, but the difference was not statistically significant (Figure 3). We have also edited the legend for Figure 3 to: “**Figure 3. *M. ulcerans* ER regulation after exposure to microaerophilic or anaerobic conditions compared to aerobic conditions.** Exposure to a microaerophilic environment for 24 hours caused significant upregulation [Mic (24H); $P=0.0009$] of ER expression (yellow bar) and transferring the bacteria back to aerobic condition led to slight downregulation in ER expression [Mic (48H), grey bar]. Exposure of *M. ulcerans* to anaerobic conditions for 24 hours led to slight ER gene upregulation [AN (24H), blue bar], but transfer back to aerobic conditions caused significant ER upregulation [AN (48H), $P=0.005$, red bar] compared to control *M. ulcerans* exposed to aerobic conditions during the entire 3-day experiment (Control bar). Error bars indicate

Standard errors. *M. ulcerans* was at 30° C for all the oxygen conditions. The updated figure 3 is:

Figure 3. *M. ulcerans* ER regulation after exposure to microaerophilic or anaerobic conditions compared to aerobic conditions. Exposure to a microaerophilic environment for 24 hours caused significant upregulation [Mic (24H); P=0.0009] of ER expression (yellow bar) and transferring the bacteria back to aerobic condition led to slight downregulation in ER expression [Mic (48H), grey bar]. Exposure of *M. ulcerans* to anaerobic conditions for 24 hours led to slight ER gene upregulation [AN (24H), blue bar], but transfer back to aerobic conditions caused significant ER upregulation [AN (48H), P=0.005, red bar] compared to control *M. ulcerans* exposed to aerobic conditions during the entire 3-day experiment (Control bar). Error bars indicate Standard errors. *M. ulcerans* was at 30° C for all the oxygen conditions.

Reviewer Comment: Line 140 Why is 25°C not mentioned? It is shown in figure 2B.

Author Response: We thank the reviewer for pointing this out and apologize for the error. This was an error on our part in submitting that original figure that had the 25°C growth data included. We did not continue with any downstream analyses at 25°C and had planned to upload a version of the figure without those data, as it did not add any important data with respect to the remainder of the manuscript data. We have included the appropriate figure 2B with those data excluded. The updated figure 2 is:

Figure 2. Effect of Oxygen (A) or temperature (B) on *M. ulcerans* growth. (A) Optical density of *M. ulcerans* when exposed to aerobic (blue line), microaerophilic (orange) and anaerobic (grey line) conditions at 30°C. Exponential *M. ulcerans* initially under aerobic conditions were exposed to their respective oxygen condition for 24 hours and then transferred back to aerobic conditions for an additional 24 hours (48 hours from initial time point). (B) Optical density of *M. ulcerans* when exposed to 30°C (blue line) and 37°C (orange line). Exponential *M. ulcerans* initially at 30°C were exposed to their respective temperature conditions for 24 hours and then exposed back to 30°C for an additional 24 hours (48 hours from initial time point).

Reviewer Comment: Line 151 Why is Mic (48H) shown before Mic (24H) on Figure 3. Should be reversed.

Author Response: We have edited figure 3 to change the ordering of the data presented with Mic(24H) first, followed by Mic (48H) (Please see above).

Reviewer comment: Line 159 I find the labeling on Figure 4 confusing. I suggest a less complicated labeling system. The descriptions are good, but the actual labels confused me.

Author Response: We thank the reviewer for the comments. We considered many ways to make the figure less confusing. In the end, we thought the best way would be to make the figure a single panel figure, rather than in two panels. This allowed us to a) box in the treatments and their conditions at each day, and b) also remove the duplicate bars of T37-O2 that were originally in each panel to only show those data once. We hope that reorganizing this figure into a single panel and including boxes and letters for each of the boxes removes confusion. The updated figure is:

Figure 4. Regulation of ER expression after *M. ulcerans* exposure to differing temperature and oxygen conditions. (A) Control conditions of *M. ulcerans* at 30⁰C and aerobic conditions from days 1-3; **(B and in blue box):** *M. ulcerans* exposed to 37⁰C and aerobic conditions on Day 2 [T37-O2 (D2)] but transferred back to 30⁰C and aerobic condition on day 3 [T30-O2 (D3)]; **(C and in red box):** *M. ulcerans* exposed to 37⁰C and microaerophilic condition on day 2 [T37-Mic (D2)] but transferred back to 30⁰C and aerobic condition on day 3 [T30-O2 (D3)]; **(D and in green box):** *M. ulcerans* exposed to 37⁰C and anaerobic condition on day 2 [T37-Anae (D2)] but transferred back to 30⁰C and aerobic condition on day 3 [T30-O2 (D3)]: Bars indicate standard error. P values indicate significance values between treatment and control. Stars indicate significance within treatments.

Reviewer Comment: Line 188 I was confused by the labeling and categories on Figure 5. Do the last two categories represent a different time frame from the first three? Why do they not represent a 3-day period? The D2 categories should be compared to the control at D2 not the control at D3 and other categories at D3.

Author Response: We thank the reviewer for the comment and hope to clarify. The Last two categories represent *M. ulcerans* gene expression from samples that were collected from flasks actively incubating at 37°C and aerobic conditions or 37°C and microaerophilic conditions (these were Day 2 of the 3 day experiment). These were compared to control samples transcriptomes incubating in 30°C and aerobic conditions not manipulated across the three day period (these control data were considered as baseline, just as with the RT-qPCR analyses) and also to the transcriptomes of samples that were collected from the same flasks that were moved back to 30°C and aerobic conditions for an additional 24 hours (Day 3 of the 3 day experiment). We have updated figure 5 to differentiate the different treatment groups. The updated figure is:

Figure 5. Fifty significantly differentially regulated genes across treatments and timepoints. T30-O2-D1-3: *M. ulcerans* grown at aerobically at 30°C for the entire 3 day study; T37-O2-D3: *M. ulcerans* grown aerobically at 37°C for 24 hours that had been moved back to control conditions for 24 hours; T37-Mic-D3: *M. ulcerans* grown microaerophilically at 37°C for 24 hours that had been moved back to control conditions for 24 hours; T37-O2-D2: *M. ulcerans* grown aerobically at 37°C for 24 hours; T37-Mic-D2: *M. ulcerans* grown microaerophilically at 37°C for 24 hours.

Reviewer Comment: Line 426-427 How many CFUs (or volume of bacterial suspension) were initially inoculated into how many mLs of M7H9?

Author Response: A 1% inoculum was added to 30mL of M7H9+OADC. We have edited the “Bacterial strains and culture” subsection of the methods to read: “A 1% inoculum of *Mycobacterium ulcerans* JKD8083 or Agy99 was inoculated into 30mL total volume Middlebrook 7H9 (M7H9) broth containing Oleic Albumin Dextrose Catalase (OADC) and incubated aerobically at 30°C for 4-6 weeks to reach exponential phase for use in this study.”

Reviewer Comment: Line 433 What spectrophotometer was used?

Author Response: We used a Thermo Scientific Genesys 20 Spectrophotometer. We have added this information into the Materials and Methods Measurement of Optical Density (OD600) subsection.

Reviewer Comment: Line 435 When was optical density used to measure *M. ulcerans* growth and when were CFU counts on serial dilutions plated on M7H10 agar used? I am not sure if both techniques were used for each experiment, but this should be clarified. (This comment corresponds to the first comment).

Author Response: We collected both sets of data at the same time that other samples and data were collected (at the specific timepoints). We reported CFU data when available, but there were some instances where our plates for CFU counts were unreadable due to fungal contamination of some of the plates used for growth analyses, giving incomplete data across all timepoints. In those cases, we instead reported OD 600 data. However, we have changed out Figure 1A to report OD600 data rather than CFU. Additionally, we have added this information into the Materials and Methods “Effect of UV exposure on *M. ulcerans* growth and mycolactone gene expression” and “Combined effect of high temperature and low oxygen in mycolactone and global gene expression” subsections.

Reviewer Comment: Line 436 How many log 10 dilutions were done, and how many mLs of each dilution was plated?

Author Response: We diluted to the 10⁻⁶ dilution, then plated 10µL of dilutions in triplicate. We have added this information to the Materials and Methods Measurement of Bacterial Growth subsection.

Reviewer Comment: Line 445 What is meant by “Briefly”?

Author Response: We have removed the word “briefly” from the body of the manuscript for clarity.

Reviewer Comment: Line 446 How was exponential growth determined?

Author Response: We have growth curve data from our laboratory from routine culture of *M. ulcerans* under aerobic conditions at 30°C showing that mid exponential phase occurs between 4-6 weeks under these growth conditions in our lab.

Reviewer Comment: Line 480 Was the supernatant removed before Trizol addition?

Author Response: We have updated the sentence to read: Bacterial cells were pelleted by centrifugation, with supernatant removed, and 1.0 mL Trizol reagent was added to the pellet and mixed thoroughly, and bead beaded.

Reviewer Comment: Line 492 Figure 6 seems to be misplaced. Should be under section starting at line 454

Author Response: We thank the reviewer for the comment; however, we respectfully disagree since figure 6 is describing the flask setup and incubation conditions for the experiments measuring the Effect of temperature and oxygen on *M. ulcerans* growth and mycolactone gene expression, and the combined temperature and oxygen effect on *M. ulcerans* growth and mycolactone gene expression. Therefore, our rationale was to reference this experimental design figure in these sections.

January 29, 2023

Dr. Heather Jordan
Mississippi State University
Biological Sciences
P.O. Box GY
Mississippi State, Mississippi 39762

Re: Spectrum04968-22R1 (Impact of Temperature and Oxygen Availability on Gene Expression Patterns of *Mycobacterium ulcerans*)

Dear Dr. Heather Jordan:

Your manuscript has been accepted, and I am forwarding it to the ASM Journals Department for publication. You will be notified when your proofs are ready to be viewed.

Sincerely,

John Attack
Editor, Microbiology Spectrum
